# Diagnostic Value of 18F-FDG PET/CT vs. Chest-Abdomen-Pelvis CT Scan in Management of Patients with Fever of Unknown Origin, Inflammation of Unknown Origin or Episodic Fever of Unknown Origin: A Comparative Multicentre Prospective Study

**DOI:** 10.3390/jcm11020386

**Published:** 2022-01-13

**Authors:** Kim-Heang Ly, Nathalie Costedoat-Chalumeau, Eric Liozon, Stéphanie Dumonteil, Jean-Pierre Ducroix, Laurent Sailler, Olivier Lidove, Boris Bienvenu, Olivier Decaux, Pierre-Yves Hatron, Amar Smail, Léonardo Astudillo, Nathalie Morel, Jonathan Boutemy, Antoinette Perlat, Eric Denes, Marc Lambert, Thomas Papo, Anne Cypierre, Elisabeth Vidal, Pierre-Marie Preux, Jacques Monteil, Anne-Laure Fauchais

**Affiliations:** 1Department of Internal Medicine, Limoges University Hospital, CEDEX, 87042 Limoges, France; eric.liozon@chu-limoges.fr (E.L.); Stephanie.Dumonteil@chu-limoges.fr (S.D.); babouvidal87@yahoo.fr (E.V.); anne-laure.fauchais@unilim.fr (A.-L.F.); 2AP-HP, Cochin University Hospital, Internal Medicine Department, Referral Centre for Rare Autoimmune and Systemic Diseases, 75014 Paris, France; nathalie.costedoat@gmail.com (N.C.-C.); namorel@orange.fr (N.M.); 3Department of Internal Medicine, Amiens University Hospital, 80054 Amiens, France; jpf.ducroix@wanadoo.fr (J.-P.D.); smail.amar@chu-amiens.fr (A.S.); 4Department of Internal Medicine, CHU Toulouse-Purpan, CEDEX, 31059 Toulouse, France; sailler.l@chu-toulouse.fr (L.S.); astudillo.l@chu-toulouse.fr (L.A.); 5Department of Internal Medicine, Groupe Hospitalier Diaconesses-Croix Saint-Simon, 75020 Paris, France; OLidove@hopital-dcss.org; 6Department of Internal Medicine, Caen University Hospital, CEDEX 9, 14033 Caen, France; bbienvenu@hopital-saint-joseph.fr (B.B.); boutemy-j@chu-caen.fr (J.B.); 7Department of Internal Medicine CHU de Rennes, 35000 Rennes, France; olivier.decaux@chu-rennes.fr (O.D.); antoinette.perlat@chu-rennes.fr (A.P.); 8Department of Internal Medicine, CHU Claude Huriez, 59000 Lille, France; pierre-yves.hatron@chru-lille.fr (P.-Y.H.); Marc.LAMBERT@CHRU-LILLE.FR (M.L.); 9Department of Infectious Diseases, CHU Limoges, CEDEX, 87042 Limoges, France; denes.eric@gmail.com (E.D.); Anne.Cypierre@chu-limoges.fr (A.C.); 10Department of Internal Medicine, Paris Diderot University, Assistance Publique-Hôpitaux de Paris, Bichat Hospital, 75018 Paris, France; Thomas.papo@bch.ap-hop-paris.fr; 11Centre d’Epidémiologie de Biostatistique et de Méthodologie de la Recherche, Limoges University Hospital, CEDEX, 87042 Limoges, France; pierre-marie.preux@unilim.fr; 12Department of Nuclear Medicine, Limoges University Hospital, CEDEX, 87042 Limoges, France; jacques.monteil@unilim.fr

**Keywords:** PET/CT, chest-abdomen-pelvis CT, FUO, IUO, EFUO

## Abstract

Fluorodesoxyglucose Positron Emission Tomography (PET/CT) has never been compared to Chest-Abdomen-Pelvis CT (CAPCT) in patients with a fever of unknown origin (FUO), inflammation of unknown origin (IUO) and episodic fever of unknown origin (EFUO) through a prospective and multicentre study. In this study, we investigated the diagnostic value of PET/CT compared to CAPCT in these patients. The trial was performed between 1 May 2008 through 28 February 2013 with 7 French University Hospital centres. Patients who fulfilled the FUO, IUO or EFUO criteria were included. Diagnostic orientation (DO), diagnostic contribution (DC) and time for diagnosis of both imaging resources were evaluated. One hundred and three patients were included with 35 FUO, 35 IUO and 33 EFUO patients. PET/CT showed both a higher DO (28.2% vs. 7.8%, *p* < 0.001) and DC (19.4% vs. 5.8%, *p* < 0.001) than CAPCT and reduced the time for diagnosis in patients (3.8 vs. 17.6 months, *p* = 0.02). Arthralgia (OR 4.90, *p* = 0.0012), DO of PET/CT (OR 4.09, *p* = 0.016), CRP > 30 mg/L (OR 3.70, *p* = 0.033), and chills (OR 3.06, *p* = 0.0248) were associated with the achievement of a diagnosis (Se: 89.1%, Sp: 56.8%). PET/CT both orients and contributes to diagnoses at a higher rate than CAPCT, especially in patients with FUO and IUO, and reduces the time for diagnosis.

## 1. Introduction

Fever of Unknown Origin (FUO), Inflammation of Unknown Origin (IUO) and Episodic Fever of Unknown Origin (EFUO) are three of the most challenging diagnostic problems for internal medicine physicians. These three entities are defined, respectively, as (I) body temperature over 38.3 °C on several occasions, with symptoms for at least three weeks and no specific diagnosis after extended hospitalization and/or outpatient investigations [1,2] (FUO), (II) the same criteria for FUO in patients with a body temperature below 38.3 °C and raised acute phase reactants (IUO), (III) recurrent or episodic fever in patients with FUO criteria in whom the fever abates spontaneously with fever-free intervals of at least 48 h (EFUO) [3,4]. The causes of FUO and IUO are generally the same in developed countries and include non-infectious inflammatory diseases (NIIDs), followed by infections, tumours and miscellaneous diseases [5]. In patients with EFUO, the major causes are miscellaneous diseases, followed by NIIDs, tumours and infections [4,6,7]. The proportion of patients without a diagnosis is growing and now represents up to 20% of patients with FUO [4,6,7,8,9,10,11] and up to 50% of patients with EFUO.

The diagnostic work-up in patients with FUO, IUO or EFUO is not well defined. Before invasive procedures, a first step of the diagnostic evaluation could include a history and physical examination, laboratory tests, blood and urine cultures, tuberculin skin test, chest X-ray and abdominal ultrasonography, as recently suggested [12]. As many patients remain without diagnosis after this first step, further investigations are needed. As a functional imaging tool, 18F-fluorodeoxyglucose positron emission tomography combined with computed tomography (PET/CT) appeared to be a helpful diagnostic tool for detecting inflammatory sites in the whole body in FUO, IUO or EFUO patients in three prospective studies [11,13,14] as well as in retrospective and small case report studies [13,15,16]. It allows for a diagnostic orientation and contributes to a diagnosis in 16% to 78% of patients in several studies [14,17,18,19,20,21,22]. A meta-analysis with a total of 1927 patients pointed out the diagnostic performance of PET/CT in the diagnosis of FUO and IUO with 84% sensitivity and specificity [16]. Data on the diagnostic contribution of PET/CT in EFUO patients is lacking [23]. However, access to PET/CT remains limited due to a lack of reimbursement for FUO in many countries [23], highlighting the importance of searching for alternative modalities such as chest-abdomen-pelvis CT (CAPCT) scan. A CAPCT scan is often preferred to PET/CT due to its accessibility and cost, which is three times less than PET/CT (respectively $385 vs. $1375 according to Medicare in 2019), especially when the clinician is confronted with a patient presenting atypical symptoms. Very few retrospective studies have explored the diagnostic contribution of standalone CAPCT in these settings, and results have been contradictory [21,24,25].

We conducted a study to compare the relative usefulness of CAPCT and PET/CT scanning in the diagnostic work up of patients with FUO/IUO/EFUO. The primary outcome of this trial was the proportion of patients with a diagnosis suggested by CAPCT or/and PET/CT. The secondary aim of this study was to assess the sensitivity and the specificity of diagnosis of these two imaging techniques for all groups of patients.

## 2. Materials and Methods

### 2.1. Study Design and Participants

We conducted a prospective, multicentre, comparative open-label study with a direct, individual-benefit trial. The trial was approved by all local ethical committees and was registered in a clinical trial site under the number NCT01200771. All patients provided written informed consent before entering the study. The study design is shown in Figure 1.

The trial was performed between May 2008 and February 2013. Patients were recruited from seven French University Hospital centres. We included all adult patients who fulfilled the FUO criteria [1,2], IUO criteria or EFUO criteria (see definitions below). Immunocompromised patients were excluded from the study, as were patients with nosocomial or HIV-associated FUO and patients using prednisone ≥10 mg/day within two weeks prior to inclusion.

### 2.2. Procedures and Diagnostic Work-Up

A first-line diagnostic work-up was needed for each patient at least two weeks before inclusion (Figure 1). After this first-line diagnostic work up, patients with signs of a potential diagnosis underwent further specific investigations for a final diagnosis. If a diagnosis had not been made, patients were registered for inclusion in the study, as were patients without any signs of a potential diagnosis after the first-line diagnostic work-up. All registered patients were hospitalized and underwent a second-line diagnostic work-up with further investigations. During this second-line diagnostic work-up, if no diagnosis was identified, patients were included in the study and all FUO and IUO patients underwent a CAPCT and PET/CT scan within a week following hospitalization. For EFUO patients, imaging was advised within three days after the onset of the fever. The interpretation of the CAPCT and PET/CT was performed blindly by the radiologist or nuclear medicine specialist so as to avoid any interpretation bias. If an oriented diagnosis sign was suggested from any investigation, appropriate examinations were performed to find a diagnosis. If not, a third-line investigation procedure was proposed to patients, based on the clinician’s judgment, with or without hospitalization, depending on the patient’s condition. A systematic visit was scheduled 6 months after inclusion to look for a possible late diagnosis. The end of follow-up was considered in case of diagnosis. Patients without a diagnosis were followed for a further six months.

### 2.3. Study Technique: 18F-FDG PET/CT and Chest-Abdomen-Pelvis CT Scan Processing

For 18F-FDG scanning, different PET/CT instruments were used in the various centres investigated. However, all centres investigated followed the same imaging process: before 18F-FDG injection, the included patients fasted for at least four hours and their blood glucose level was checked to ensure it was less than 180 mg/dL; scans were performed one hour after injection, with an average injection time equal to 74.3 (14.3) minutes; a non-contrast low-dose-radiation CT scan was performed first followed by a PET scan encompassing the same imaging field; patients were in the supine position. PET data were reconstructed by iterative methods with standard software shipped with the system and fused in PET/CT slices for evaluation. The data were corrected for attenuated and scattered photons. PET scans were evaluated by a nuclear medicine specialist according to each centre’s standard practice, blinded to the patient’s case report form and CAPCT scan imaging data. PET/CT exams were regarded as pathological if moderate to high focal tracer uptake was detected in addition to areas of physiological tracer uptake (kidney, brain, heart, urinary bladder, intestinal smooth muscle, liver, spleen and testes). All PET/CT scans were blindly re-evaluated by a reference nuclear medicine specialist. In case of discordance between the first and the second PET/CT interpretation, a final statement on the PET/CT results was reached by consensus between the reference nuclear medicine specialist and the reference internist, with only an understanding of the patient’s medical history and clinical chart this time.

CAPCT scans, defined as a contrast-enhanced CT of the thorax, abdomen and pelvis, were performed for all the patients and were evaluated by radiologists who were blind to the patient’s medical history and PET/CT results.

### 2.4. Outcomes and Results Interpretation

The primary endpoint was the proportion of patients for whom a PET/CT or CAPCT scan contributed to diagnosis. Results of both techniques were interpreted as follows: (I) diagnostic orientation (DO) was considered when the PET/CT or CAPCT scan showed abnormalities that could not be explained, respectively, by physiological tracer uptake or anatomical variations, (II) diagnostic contribution (DC) was considered when abnormal results from an imaging technique were referred to an organ or tissue for which further conventional techniques (i.e., biopsy) led to a diagnosis (true positive), (III) abnormal results were regarded as false positives if the abnormalities detected were inconsistent with subsequent test, unrelated to the final diagnosis or had an inconsistent outcome, (IV) a normal imaging technique was regarded as a true negative if the cause was not detectable using this technique and if no diagnosis was made despite the third-line diagnostic work-up and at the end of follow-up, (V) an imaging technique was regarded as a false negative if another investigation led to a diagnosis during the third-line investigation period and before the end of follow-up. A review committee was established to obtain a consistent interpretation of the results.

For a secondary endpoint, we used these definitions of result interpretations to assess sensitivity and specificity for diagnosis of each imaging technique.

### 2.5. Statistical Analysis

All analyses were conducted using R software (Version 3.2.2). Continuous quantitative variables were represented as means and standard deviation (SD), qualitative variables as percentages. For analytic study, Pearson’s χ2 test was used to compare qualitative variables between groups of patients. To compare quantitative variables between groups, the appropriate test was used (Student’s *t*-test, Wilcoxon or Kruskal–Wallis).

After the univariate analysis of the diagnostic predictive factors, the variables with a *p*-value less than 0.20 were included in a logistic regression model. Quantitative variables that verified the linearity hypothesis of the logit were integrated without modification. The initial multivariate model was simplified by a backward-stepwise elimination method, such that the final model included only variables significantly associated with the variable diagnosis. Model calibration was assessed using Pearson residual and deviance residual tests. The ROC curve analysis of data was performed and the area under the ROC curve (AUC) equal to 1.0 was considered to be the most reliable detection indicator. A *p*-value < 0.05 was considered to be significant. Agreement between the first and second analysis of PET/CT by the reference nuclear medicine specialist was verified using the Cohen’s Kappa coefficient test.

## 3. Results

### 3.1. Patients and Diagnosis

We included 103 patients, for whom the mean (SD) age was 58.2 (16.7) years old and the mean (SD) duration of symptoms (delay between the onset of disease and patient’s inclusion) was 16.8 (5.8) months. Each group of patients represented one-third of the cohort (FUO n = 35; EFUO n = 33; IUO n = 35). Table 1 shows the clinical and biological findings of the FUO, EFUO and IUO groups. The mean (SD) duration of fever before inclusion in patients with FUO and EFUO was 39.9 (4.3) weeks. The most common clinical signs were arthralgia (49.5%), sweating (46.1%), chills (45.1%) and myalgia (34.0%). No significant differences were found between groups regarding medical history findings (Table A1) but differences were observed regarding the duration of illness and both clinical and biological presentations.

A diagnosis was made in 58 patients (56.3%) (Table 2). NIID (systemic vasculitis, rheumatic diseases, adult-onset Still’s disease (AOSD) and autoimmune diseases) accounted for 35 diagnoses (60.3%). In this subgroup, the main cause was systemic vasculitis, representing 14.6% of all patients and 25.9% of all diagnoses. Most patients with vasculitis had giant cell arteritis (GCA) and of the ten patients with rheumatic diseases (17.2% of patients with diagnosis), six had isolated polymyalgia rheumatica. In the FUO and IUO groups, systemic vasculitis predominated, while rheumatic diseases were more prevalent in the IUO group and miscellaneous diseases represented the most common disease in the EFUO group.

Patients without diagnosis (n = 45/103, 43.7%) mainly had EFUO (n = 19/45, 42.2%), while there were 15 patients with IUO (33.3%) and 11 patients with FUO (24.4%).

### 3.2. Diagnostic Orientation (DO) and Contribution (DC) of PET/CT and CAPCT Scan

Cohen’s Kappa test was performed to evaluate the agreement between the first and second analysis of PET/CT by the reference nuclear medicine specialist. With a result of 0.7 (*p* < 0.001), this test mitigated the reading and centre bias effects. The PET/CT provided diagnostic orientation (DO) in 29 patients (28.2 % of total patients and 40% of diagnosed patients (23/58), Figure 2), especially in the FUO and IUO groups, while the CAPCT scan did so in eight patients (28.2% vs. 7.8% of all patients, *p* = 0.0003). DO on the PET/CT significantly reduced the time to diagnosis compared to a normal PET/CT, with 3.8 (4.6) months vs. 17.6 (34.1) months, *p* = 0.023, especially for patients with FUO or IUO: FUO = 6.0 (15.1) months, EFUO = 30.5 (49.7) months, IUO = 7.0 (6.8) months, *p* = 0.0007. DO on the CAPCT scan did not influence the delay in diagnosis compared with normal CAPCT results. The delay was slightly reduced for patients with PET/CT-related orientation diagnosis compared to patients with CAPCT scan-related orientation, with 2.2 (1.6) months vs. 3.8 (4.9) months, *p* = 0.25.

DC was 19.4% (20 patients) for PET/CT vs. 5.8% (6 patients) for CAPCT (*p* < 0.0001) (Figure 2).

Finally, the PET/CT contributed to diagnosis with 36.4% sensitivity and 81.2% specificity, while the CAPCT contributed to diagnosis with 10.5% sensitivity and 95.6% specificity.

If we consider the group of FUO and IUO, thus excluding EFUO, we obtain a sensitivity of 45.2% and a specificity of 75.0% for PET/CT, while CT allows for a sensitivity of 9.3% and a specificity of 92.6%.

Of the 20 patients with the PET/CT DC, 12 (60.0%) had NIIDs, four (20.0%) had tumours, two (10.0%) had infections and two (10.0%) had miscellaneous diseases. Of the six patients with a DC on the CAPCT, three had malignant lymphoma, two had infections, and one had mesenteric panniculitis. All patients with a positive CAPCT scan also had a positive PET/CT except for the patient with tuberculous lymphadenitis, who presented with EFUO. In this patient, a diagnosis was made six months later with retroperitoneal node biopsy.

### 3.3. Diagnoses Not Related to the PET/CT and/or CAPCT Scan Results

After a follow-up of at least six months, 58 patients had a final diagnosis, and 45 patients did not (Figure 2, Figure 3, Figure 4 and Figure 5). Regarding the diagnostic performance of both imaging resources, there were 11 false positive patients (nine with PET/CT vs. two with CAPCT) (Figure 6). A diagnosis was made in four of the patients. Thirty-five patients with a negative PET/CT vs. 51 patients with a negative CAPCT (false negative) received a diagnosis through other complementary investigations. No diagnosis could be made in 39 patients with a negative PET/CT vs. 44 patients with a negative CAPCT (true negative).

### 3.4. Predictive Factors for Diagnosis

Arthralgia (OR = 4.90, 95%CI = [1.92–13.38], *p* = 0.0012), DO of PET/CT (OR = 4.09, 95%CI = [1.36–13.88], *p* = 0.016), CRP > 30 mg/L (OR = 3.70, 95%CI = [1.16–13.28], *p* = 0.033), and chills (OR = 3.06, 95%CI = [1.18–8.49], *p* = 0.0248) were independently associated with achieving a diagnosis. The model reached a sensitivity of 89.1%, a specificity of 56.8% and an AUC = 0.81.

## 4. Discussion

This study is the first to demonstrate, prospectively, that PET/CT is superior to CAPCT in contributing to the final diagnosis of FUO, IUO and EFUO patients and reducing the delay of diagnosis. We found that NIIDs represented the most common diagnosis, especially in patients with FUO and IUO. EFUO patients were more likely to have miscellaneous diseases, which constitutes a distinctive subset of FUO patients, with different clinical and biological profiles.

The low proportion of infectious and malignancy etiologies, of less than 8.0%, as well as a trend toward a high proportion of NIIDs and patients without diagnosis in the present study, is consistent with what previous studies have found [11,13,14]. Better access to healthcare combined with advances in technology, both microbiological and morphological, can explain this evolution. On the other hand, the application of recently upgraded diagnostic criteria for FUO and the stringent stratification of diagnostic procedures for patients included in the present study likely contributed to a more rigorous selection of difficult cases, which may explain the increased proportion of patients without a final diagnosis. Lastly, the increasing number of patients with NIIDs could be related to a better identification of systemic diseases, including their atypical forms.

This study was the first prospective protocol to include three groups of patients usually found in the clinical practice of general medicine physicians. We showed that EFUO patients differed from FUO and IUO patients in several aspects, including clinical and biological presentations, diagnostic findings and the DC of PET/CT. In this subset of patients, the diagnosis rate was lower than in other subgroups, and the DO and DC of PET/CT was less than 10.0%. Inadequate timing of realization, amongst other reasons, might explain the low diagnostic performance of PET/CT in these patients with episodic inflammation.

Three retrospective studies compared the contribution of CAPCT and PET/CT in the management of patients with FUO [24,25] and IUO [21]. In these studies, PET/CT provided a higher DC than CAPCT. It is worth noting that the PET/CT recognized diagnoses identified by CAPCT in all cases but one. The only analogous patient had Still’s disease with detectable spleen enlargement, and received corticosteroids before the PET/CT. Likewise, in the present study, one single patient with EFUO related to lymph node tuberculosis had a positive CAPCT and false negative PET/CT. The absence of FDG uptake in pathological granuloma may explain these rare discrepancies.

The DC of PET/CT was lower (19.4%) in the present study than in other prospective studies [11,13,18]. There could be several reasons for this, such as, (I) we performed two steps of diagnostic work-up before including patients, thereby eliminating easy-to-detect causes that may overstate the DC of CAPCT and PET/CT; (II) the inclusion of an EFUO group likely reduced the contribution of PET/CT, since both the diagnosis rate and the DC of PET/CT were found to be low in this group [9,10], (III) the multicentre study design reduced the problems inherent to monocentric studies, such as recruitment biases, which would overestimate the DC of PET/CT. In this respect, a PET/CT contributed to a diagnosis in 27.0% of patients with FUO or IUO included in the present study, which is in agreement with the 33.0% contribution observed in a multicentre prospective study [13,18] (IV) finally, the high proportion of NIIDs (45/58, 77.5%) where PET/CT is often unrevealing, unlike CAPCT, may have reduced the DC rate and increased the false-positive rate of this imaging tool.

Therefore, the choice of PET/CT over CAPCT in the diagnostic work-up of patients with FUO/IUO merits further discussion. The radiation dose from CT is a major concern when using PET/CT for patients with FUO, IUO or EFUO. The effective dose with the administration of 185 MBq 18F-fluorodeoxyglucose is estimated to be 3.5 mSv [26], while the effective dose from the CT component could range from 1 to 20 mSv. Radiation exposure appears to be higher with PET/CT since it incorporates the effective dose from the CT scan component to that from the tracer. However, using a non-contrast low-dose-radiation CT scan when performing PET/CT may contribute to reduced radiation exposure as well as using new hybrid imaging as Whole body PET/MRI, which appears to provide similar diagnostic usefulness to PET/CT [27,28].

Indeed, the inclusion criteria in several studies were different and the high proportion of patients without a diagnosis and with a diagnosis of NIID did not allow for a correct assessment of the usefulness of PET/CT in this setting. On the other hand, we have shown that a PET/CT considerably reduced the time to diagnosis and management of these patients. We could suggest that a PET/CT would be more relevant in the diagnostic process of these patients if we could identify a patient profile for whom PET/CT would contribute greatly to the diagnosis.

In our study, we showed that the presence of chills, arthralgia, a diagnostic orientation to PET/CT and a CRP >30 mg/L were predictive factors of diagnosis. This is in agreement with the findings of Schönau et al. [11] of an age >50 years and a CRP >30 mg/L being predictive factors for DC of PET/CT.

Our study has several limitations that need to be emphasized. The inclusion of an EFUO group may have added complexity to the study design and limited the results. However, making this choice allowed us to provide results closer to the daily medical practice of general medicine. Second, the small sample size of each group of patients may have decreased the accuracy of the results.

## 5. Conclusions

We showed that PET/CT is more useful for establishing a diagnosis and reducing time diagnosis, especially in patients with FUO and IUO. Efforts should be made to secure reimbursement for FDG-PET imaging for the workup of FUO on a routine basis, so as to minimize human suffering and to avoid unnecessary and costly procedures and interventions.

## Figures and Tables

**Figure 1 jcm-11-00386-f001:**
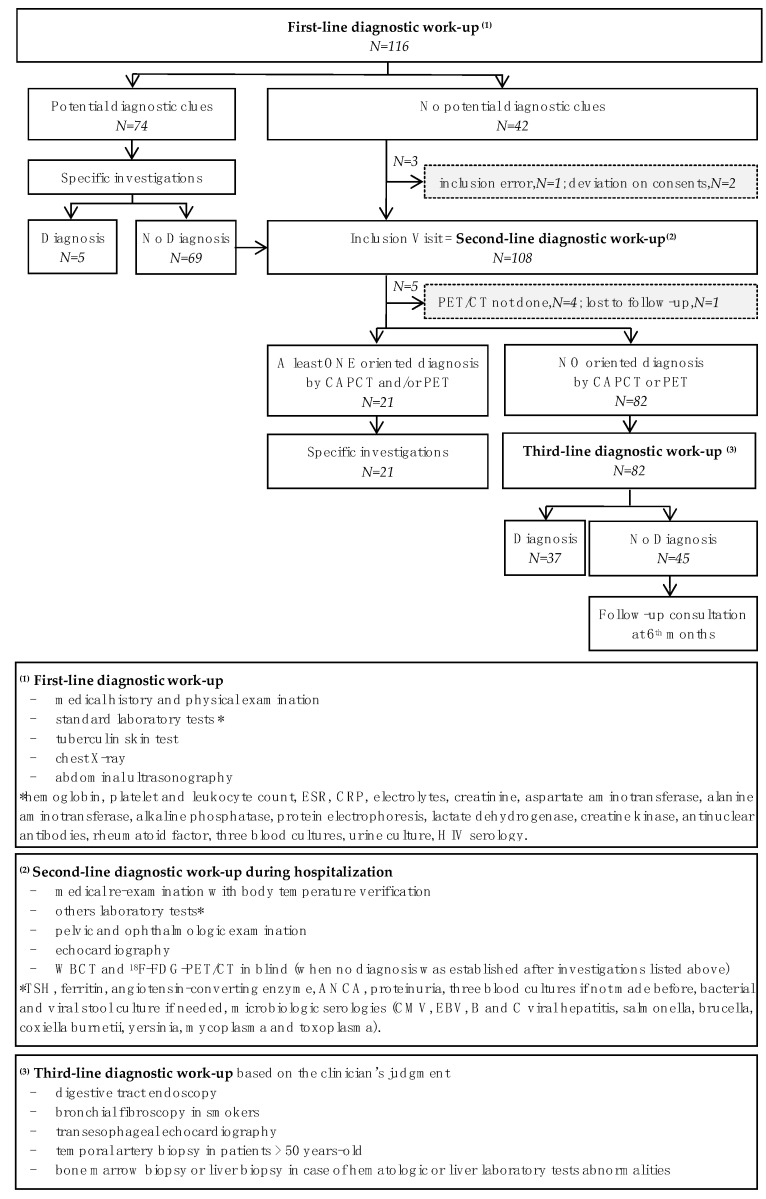
Diagnostic work-up of FUO-TEP study. Legends Figure 1: ANCA: antineutrophil cytoplasmic antibodies; CAPCT: Chest-abdomen-pelvis CT, CMV: Cytomegalovirus; CRP: C-reactive protein; EBV: Epstein–Barr virus; ESR: Erythrocyte sedimentation rate; PET/CT: 18F-fluorodeoxyglucose positron emission tomography combined with Computed Tomography; TSH: thyroid-stimulating hormone.

**Figure 2 jcm-11-00386-f002:**
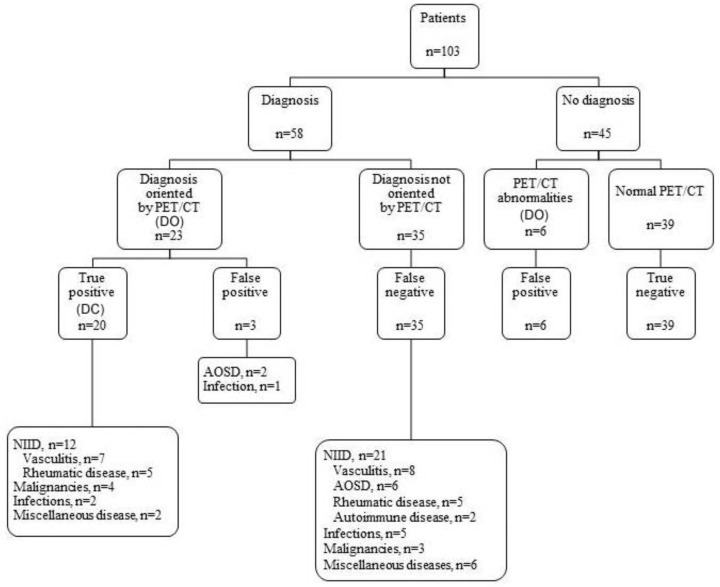
Distribution of diagnosis and helpfulness of PET/CT. AOSD: Adult onset Still’s disease; DC: diagnostic contribution; DO: diagnostic orientation; NIID: non-infectious inflammatory diseases; PET/CT: 18F-fluorodeoxyglucose positron emission tomography combined with Computed Tomography.

**Figure 3 jcm-11-00386-f003:**
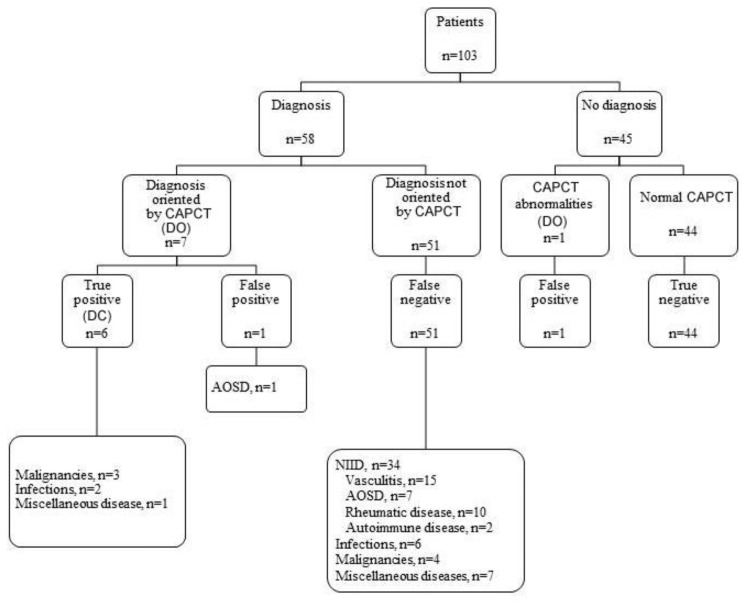
Distribution of diagnosis and helpfulness of CAPCT Legends Figure 3: AOSD: Adult onset Still’s disease; DC: diagnostic contribution; DO: diagnostic orientation NIID: non-infectious inflammatory diseases; CAPCT: Chest-abdomen-pelvis CT.

**Figure 4 jcm-11-00386-f004:**
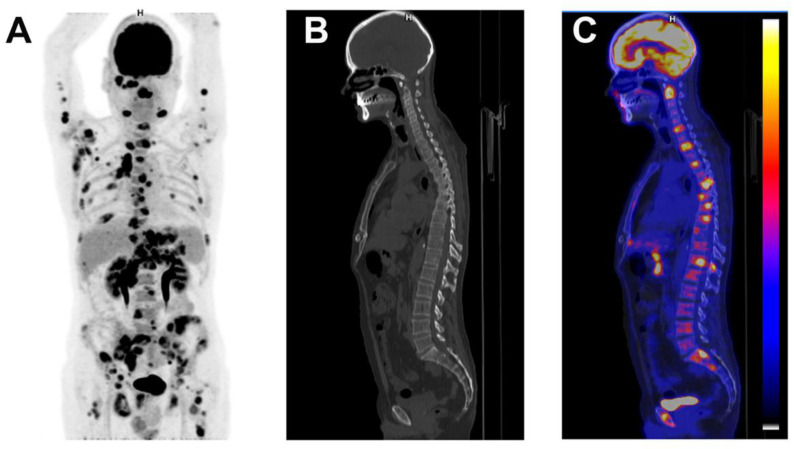
Patient with diffuse large B cell lymphoma. A 52-year-old man was admitted with 7 week’s duration of FUO, sweating, myalgia and lower back pain. Physical examination showed axillary lymph nodes. Abdominal examination did not show hepatosplenomegaly. Laboratory testing showed a CRP of 132 mg/L with normal leucocytes. ASAT was 43 U/L, ALAT 67 U/L, alkaline phosphatase 155 U/L and. Angiotensin converting enzyme was normal. Blood cultures were negative as were bartonella, Q Fever, mycoplasma, chlamydia, HVC, HBV and toxoplasmosis serologies with past EBV and CMV immunity. The CAPCT scan performed 2 weeks before hospitalization was normal. Coronal maximum intensity projection FDG-PET (**A**), sagittal low-dose CT (**B**) and fused FDG-PET/CT (**C**) demonstrated intensive FDG uptake in axillary, submandibular, mediastinal, para-aortic, epigastric and inguinal lymph nodes and in bones. A submandibular lymph node confirmed the diagnosis of diffuse large B cell lymphoma.

**Figure 5 jcm-11-00386-f005:**
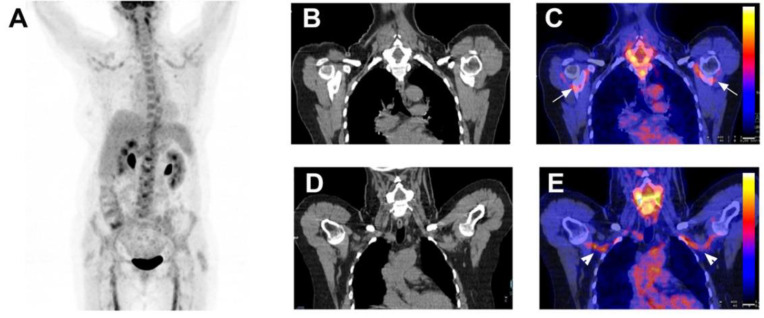
A 64-year-old female patient with giant cell arteritis associated with polymyalgia rheumatica. A 64-year-old female patient was admitted with asthenia, headaches, shoulder, knee and elbow pain for 4 months. She had no fever. On physical examination, there was no sign of jaw claudication nor scalp tenderness. The temporal arteries were normal. White blood cells were normal. CRP was 35 mg/L. Temporal artery biopsy was negative. Coronal maximum intensity projection FDG-PET (**A**), coronal low-dose CT (**B**,**D**) and fused FDG-PET/CT (**C**,**E**) revealed FDG uptake in shoulders ((**C**) white arrowheads) and subclavian and axillary arteries ((**E**) white arrowheads). The diagnosis of giant cell arteritis associated with polymyalgia rheumatica was established. Her symptoms resolved with corticosteroids.

**Figure 6 jcm-11-00386-f006:**
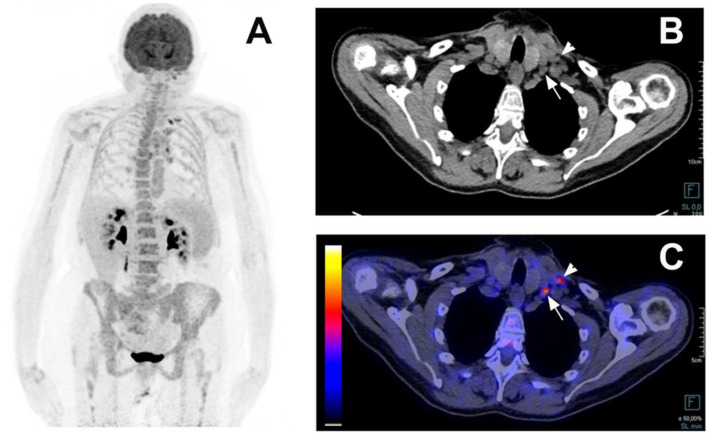
This examples a false positive PET/CT in a 70-year-old woman with Still disease. A 70-year-old woman presented with an episodic fever for 4 years with fatigue and a weight loss of 5 kg. She presented with one episode per year for the first two years and two episodes in the 4th year. During the fever episodes, she complained of joint pain in her wrists, knees and ankles and sometimes of an erythematous maculo-papular rash on her back. On physical examination, there was no synovitis of the painful joints, no skin rash, no adenopathy and no spleen or liver enlargement. Leukocytes were 9.65 G/L, CRP was 164 mg/L, alkaline phosphatase 116 U/L, ferritin was 3000 ng/mL. Blood cultures were negative as were Q Fever, HVC, HBV serologies with past toxoplasmosis EBV and CMV immunity. Anti-nuclear and anti-CCP antibodies were negative. The CAPCT scan performed during hospitalization showed mediastinal lymph nodes. Coronal maximum intensity projection FDG-PET (**A**), axial low-dose CT (**B**), and axial fused FDG-PET/CT (**C**) showed increased FDG uptake in left subclavian (white arrowheads) and mediastinal lymph nodes. In the hypothesis of lymphoma, a left subclavian lymph node biopsy was performed and showed a reactive adenitis. The diagnosis of adult onset Still disease was maintained as she met the Fautrel criteria with a glycosylated ferritin level at 8%.

**Table 1 jcm-11-00386-t001:** Clinical and biological findings of the FUO, EFUO and IUO groups.

	FUO n = 35	IUO n = 35	EFUO n = 33	Total n = 103	*p* Value
Variables	n (%) or mean (SD)				
Sex: Male/Female	20/15	18/17	15/18	53/50	0.6285
Age (years)	59.3 (16.7)	65.1 (13.6)	49.8 (16.6)	58.2 (16.7)	0.0005
Duration of symptoms (months)	7.7 (18.0)	9.4 (12.9)	34.5 (55.9)	16.8 (35.8)	<0.0001
Maximum fever (°C)	39.5 (0.7)	38.5 (0.3)	39.6 (0.6)	39.4 (0.7)	0.0002
Duration of fever (weeks)	24.1 (12.1)	4.4 (1.2)	67.2 (31.3)	39.9 (14.3)	0.0309
Arthralgia (%)	15 (42.8)	22 (62.8)	14 (42.4)	51 (49.5)	0.1513
Sweating (%)	20 (57.1)	8 (22.9)	19 (59.4)	47 (46.1)	0.0030
Chills (%)	19 (54.3)	4 (11.4)	23 (71.9)	46 (45.1)	<0.0001
Myalgia (%)	17 (48.6)	9 (25.7)	9 (27.3)	35 (34.0)	0.0801
Cutaneous signs (%)	13 (37.1)	2 (5.7)	10 (30.3)	25 (24.3)	0.0056
Headaches (%)	8 (22.9)	7 (20.0)	7 (21.2)	22 (21.4)	0.9581
Arthritis (%)	7 (20.6)	5 (14.3)	4 (12.1)	16 (15.7)	0.6106
Transit disorders (%)	2 (5.9)	2 (5.7)	10 (30.3)	14 (13.7)	0.0035
Heart murmur (%)	4 (11.8)	9 (25.7)	0 (0.0)	13 (12.7)	0.0063
Lymph nodes (%)	5 (14.3)	2 (5.7)	4 (12.1)	11 (10.7)	0.4834
Hepatomegaly (%)	4 (11.8)	1 (2.9)	1 (3.0)	6 (5.9)	0.2110
Abdominal pain (%)	1 (2.9)	1 (2.9)	1 (3.0)	3 (2.9)	0.9988
White blood cells	9.2 (4.3)	9.2 (3.2)	9.5 (3.9)	9.3 (3.8)	0.2779
Haemoglobin	11.4 (1.5)	11.1 (1.8)	11.8 (2.2)	11.4 (1.8)	0.0741
Platelets	347.1 (191.8)	394.7 (177.2)	306.6 (117.5)	350.7 (168.8)	0.0329 *
ESR	74.3 (34.5)	87.6 (35.0)	58.6 (35.2)	73.7 (36.4)	0.0097 *
CRP	111.4 (81.2)	90.6 (54.5)	61.2 (64.3)	88.8 (70.2)	0.0201 #
AST	33.1 (3.8)	22.7 (1.7)	29.0 (3.4)	28.3 (18.2)	0.0327 #
ALT	37.9 (24.6)	20.5 (12.7)	30.7 (19.5)	29.8 (20.8)	0.0020 #
LDH	446.6 (258.1)	320.1 (139.5)	374.3 (156.3)	382.0 (199.0)	0.0253 +
Albuminemia	32.0 (6.6)	31.6 (5.4)	35.3 (5.7)	32.9 (6.1)	0.0346 ** 0.0193 *

ALT: Alanine transaminase; AST: Aspartate transaminase; CRP: C-reactive protein; EFUO: episodic FUO; ESR: erythrocyte sedimentation rate; FUO: fever of unknown origin; IUO: inflammation of unknown origin; LDH: lactate dehydrogenase. Significant difference: * between EFUO and IUO groups; ** between FUO and EFUO groups; + between FUO and IUO groups; # between each group.

**Table 2 jcm-11-00386-t002:** Etiologies of FUO, EFUO and IUO groups.

	FUO n = 35	IUO n = 35	EFUO n = 33	Total n = 103	*p* Value
Variables	n (%)				
Non-diagnosis	11 (31.4)	15 (42.9)	19 (57.6)	45 (43.7)	
Diagnosis	24 (68.6)	20 (57.1)	14 (42.4)	58 (56.3)	
	n (% of diagnoses)				
NIID	15 (62.5)	16 (80.0)	4 (28.6)	35 (60.3)	0.0101
Systemic vasculitis	7 (29.2)	8 (40.0)	0 (0.0)	15 (25.9)	0.0263
Rheumatic disease	1 (4.2)	8 (40.0)	1 (7.1)	10 (17.2)	0.0008
Still’s disease	5 (20.8)	0 (0.0)	3 (21.4)	8 (13.8)	0.0779
Autoimmune diseases	2 (8.3)	0 (0.0)	0 (0.0)	2 (3.4)	0.1379
Infections	3 (12.5)	1 (5.0)	4 (28.6)	8 (13.8)	0.3530
Miscellaneous diseases	1 (4.2)	1 (5.0)	6 (42.8)	8 (13.8)	0.0014
Malignancy	5 (20.8)	2 (10.0)	0 (0.0)	7 (12.1)	0.0617

Miscellaneous diseases included: recurrent pericarditis, Erdheim-Chester disease, diverticular fistula, Muckle-Wells syndrome, Encephalitis causing IUO, morbid obesity, mesenteric panniculitis, Familial Mediterranean Fever.

## Data Availability

The datasets used and analyzed during this study are available from the corresponding author upon reasonable request.

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
