# Peer review of "Diagnostic Value of 18F-FDG PET/CT vs. Chest-Abdomen-Pelvis CT Scan in Management of Patients with Fever of Unknown Origin, Inflammation of Unknown Origin or Episodic Fever of Unknown Origin: A Comparative Multicentre Prospective Study"

_jcm, 2022, doi:10.3390/jcm11020386_

Round 1

Reviewer 1 Report

Congratulations for the work done. Some minor revisions are advisable.
A linguistic review might be useful.
Roman numerals in line 50, 53 and 54 must be capitalized: (I), (II), (III). Same thing for Roman numerals between line 149 and line 162. Same thing for line 339 and later in the discussion.
Line 188: Does the number (16.7) indicate the standard deviation? if yes, indicate accordingly. Ditto online 193.
Figure 1: it doesn't seem clear to me; is it a flow-chart? if so, it is essential to insert arrows between the levels to clarify the consecutiveness of the diagnostic steps.
Line 184-186: eliminate!
Line 218: not clear to me, please rephrase the sentence.
Line 312: Did you mean "etiology"? probably the term is more appropriate in this context.

Author Response

Reviewer 1:

The authors thank Reviewer 1 for his time and his interest in our study. 

Roman numerals in line 50, 53 and 54 must be capitalized: (I), (II), (III). Same thing for Roman numerals between line 149 and line 162. Same thing for line 339 and later in the discussion.

Line 188: Does the number (16.7) indicate the standard deviation? if yes, indicate accordingly. Ditto online 193.

Roman numerals have been capitalized in introduction and discussion sections and standard deviation has been indicated in the results section

Figure 1: it doesn't seem clear to me; is it a flow-chart? if so, it is essential to insert arrows between the levels to clarify the consecutiveness of the diagnostic steps.

Figure 1 has been modified to appear as a flow chart

Line 184-186: eliminate!

These lines have been removed

Line 218: not clear to me, please rephrase the sentence.

Sentence revised (Cohen’s Kappa test was performed to evaluate agreement between the first and second analysis of PET/CT by the reference nuclear medicine specialist . With a result of 0.7 (P<0.001), this test allowed the elimination of reading and centre bias effects.) 

Line 312: Did you mean "etiology"? probably the term is more appropriate in this context.

Corrected to “etiologies”

Reviewer 2 Report

I have the following comments on this overall interesting and well written manuscript:

1) In the Introduction and Discussion sections, please illustrate and discuss in more detail the actual reasons why, despite the advantages of PET/CT over contrast-enhanced CAPCT, the latter tends to be preferred in clinical practice. Such reasons may include e.g., earlier appointments for CAPCT than for PET/CT examinations in some centers, perception by some clinicians that CAPCT may provide more information in patients with comorbitidies or unclear clinical scenarios, etc.
2) In the same sections, another issue that should be taken into account and discussed is the potentially large radiation dose that could unnecessarily be delivered to patients undergoing CAPCT instead of PET/CT. To the latter regard, a comparison of the average ionizing radiation exposure between contrast-enhanced (eventually multiphase) CAPCT and PET/CT in the workup of patients with FUO/IUO/EFUO should also be made. Moreover, a brief mention should be made to alternative imaging modalities, such as whole body MRI. Relevant literature references supporting those points that should be added include, but are not limited to, the following:

a) Bastiani L et al, JAMA Netw Open. 2021 Oct 1;4(10):e2128561. doi: 10.1001/jamanetworkopen.2021.28561

b) Damasio MB et al, Radiol Med. 2016 May;121(5):454-61. doi: 10.1007/s11547-015-0619-9
c) Pelosi E et al, Radiol Med. 2011 Aug;116(5):809-20. doi: 10.1007/s11547-011-0649-x

2) Results. Please delete the following sentence at the beginning of the section: "This section may be divided by subheadings. It should provide a concise and precise description of the experimental results, their interpretation, as well as the experimental conclusions that can be drawn".
3.1. Patients and diagnosis. Why were 103 patients included? Was some sort of apriori sample size calculation performed before patient enrollment (also given that this was a prospective study)? More generally, was statistical power analysis performed to ensure that the sample size was large enough to draw meaningful conclusions from the data?

3) Figure 1 is unclear (I would expect that it should appear as a flow diagram, somewhat similar to Fig. 2). Please clarify and/or correct.

Author Response

Reviewer 2:

The authors thank Reviewer 2 for his time and his interest in our study.

I have the following comments on this overall interesting and well written manuscript:

In the Introduction and Discussion sections, please illustrate and discuss in more detail the actual reasons why, despite the advantages of PET/CT over contrast-enhanced CAPCT, the latter tends to be preferred in clinical practice. Such reasons may include e.g., earlier appointments for CAPCT than for PET/CT examinations in some centers, perception by some clinicians that CAPCT may provide more information in patients with comorbitidies or unclear clinical scenarios, etc.

Details have been added in the introduction section (CAPCT scan is often preferred to PET/CT due to its accessibility and cost which is three times less than PET/CT (respectively $385 vs $1375 according to Medicare in 2019), especially when the clinician is confronted with a patient presenting atypical symptoms)

In the same sections, another issue that should be taken into account and discussed is the potentially large radiation dose that could unnecessarily be delivered to patients undergoing CAPCT instead of PET/CT. To the latter regard, a comparison of the average ionizing radiation exposure between contrast-enhanced (eventually multiphase) CAPCT and PET/CT in the workup of patients with FUO/IUO/EFUO should also be made. Moreover, a brief mention should be made to alternative imaging modalities, such as whole body MRI. Relevant literature references supporting those points that should be added include, but are not limited to, the following:

  1. a) Bastiani L et al, JAMA Netw Open. 2021 Oct 1;4(10):e2128561. doi: 10.1001/jamanetworkopen.2021.28561
  2. b) Damasio MB et al, Radiol Med. 2016 May;121(5):454-61. doi: 10.1007/s11547-015-0619-9
    c) Pelosi E et al, Radiol Med. 2011 Aug;116(5):809-20. doi: 10.1007/s11547-011-0649-x

We provided information in the discussion section about radiation exposure and the value of whole body PET/IRM in this context (Radiation dose from CT is a major concern when using PET/CT for patients with FUO, IUO or EFUO. Effective dose with administration of 185 MBq 18F-fluorodeoxyglucose is es-timated to be 3.5 mSv [26] while effective dose from the CT component could range from 1 to 20 mSv. Radiation exposure appears to be higher with PET/CT since it adds effective dose from the CT scan component to that from the tracer. However using a non-contrast low-dose-radiation CT scan when performing PET/CT may contribute to reduce radiation exposure as well as using new hybrid imaging as Whole body PET/MRI which appears to provide similar diagnostic usefulness to PET/CT [27-28]). We also added new references including b) Damasio MB et al, Radiol Med. 2016 May;121(5):454-61. 

Results. Please delete the following sentence at the beginning of the section: "This section may be divided by subheadings. It should provide a concise and precise description of the experimental results, their interpretation, as well as the experimental conclusions that can be drawn".

Sentence has been deleted

Patients and diagnosis. Why were 103 patients included? Was some sort of apriori sample size calculation performed before patient enrollment (also given that this was a prospective study)? More generally, was statistical power analysis performed to ensure that the sample size was large enough to draw meaningful conclusions from the data?

The number of patients required for the study is calculated on the assumption that PET/CT significantly improves diagnostic cost-effectiveness: it would be contributory, depending on the different studies, in 26 to 69% of cases (bearing in mind that this cost-effectiveness is minimised in the literature by the patient inclusion criteria: cf. the number of diagnoses made in these studies is much lower than the results of studies on "classic" FUO). CT scan and gallium scintigraphy are contributory in 15-20% of cases for the majority of teams. PET/CT should therefore improve the cost-effectiveness of the conventional strategy by at least 20%: the number of patients needed, considering the patient as his own control, with a power of 80% and an alpha risk of 5%, is 91 patients. We therefore decided to include 100 patients.

Figure 1 is unclear (I would expect that it should appear as a flow diagram, somewhat similar to Fig. 2). Please clarify and/or correct

Figure 1 has been modified to appear as a flow chart

Reviewer 3 Report

The authors present the usefulness of PET / CT versus CTTAP in patients with fever of unknown origin, inflammation of unknown origin, and intermittent fever of unknown origin.
The number of patients is small, since although there are 103 they must be distributed in the three referred groups, this means that the results must be interpreted with caution. The concepts for the techniques of diagnostic orientation and diagnostic contribution are used, but not parameters oriented to the management of the patient. PET / CT examination shows a higher number of false positives, so its use instead of first-line CT may be questionable. It is necessary to consider the accessibility to the technique and the optimization of resources. A larger study would be necessary to be able to define which patient profile will benefit from PET / CT in the first image line and in which CT should be used first.
One of the cases presented in the article in the examination showed palpable adenopathy, which in that case would probably be the lesson technique for diagnosis would be an echo-guided FNAB and once the diagnosis of lymphoma was made, then perform PET for staging.
The optimization of material resources and justification of the ionizing radiation received by the patient with the best results for the treatment of the patient is what should prioritize the diagnostic imaging algorithms

Author Response

Reviewer 3 :

The authors present the usefulness of PET / CT versus CTTAP in patients with fever of unknown origin, inflammation of unknown origin, and intermittent fever of unknown origin.
The number of patients is small, since although there are 103 they must be distributed in the three referred groups, this means that the results must be interpreted with caution. The concepts for the techniques of diagnostic orientation and diagnostic contribution are used, but not parameters oriented to the management of the patient. PET / CT examination shows a higher number of false positives, so its use instead of first-line CT may be questionable. It is necessary to consider the accessibility to the technique and the optimization of resources. A larger study would be necessary to be able to define which patient profile will benefit from PET / CT in the first image line and in which CT should be used first.

We thank Reviewer 3 for his time and his interest in our study. We do agree that results of our study, because of the small sample size of each group of patients, must be interpreted with caution. However, our results are difficult to compare with previous studies due to the heterogeneity of the inclusion criteria between studies. We have chosen very restrictive inclusion criteria considering the technological advances in microbiology and imaging that currently allow us to refine the diagnosis of a patient with FUO, IUO or EFUO. This may explain the high number of patients without a diagnosis.

Despite a high number of false positives, the value of PET/CT in this context outweighs that of CT scan and seems to shorten the time to diagnosis and the management of these patients.

One of the cases presented in the article in the examination showed palpable adenopathy, which in that case would probably be the lesson technique for diagnosis would be an echo-guided FNAB and once the diagnosis of lymphoma was made, then perform PET for staging.
The optimization of material resources and justification of the ionizing radiation received by the patient with the best results for the treatment of the patient is what should prioritize the diagnostic imaging algorithms

Concerning the patient with palpable adenopathy, in real life, it is obviously necessary to give priority to clinical evaluation at all times in the management of these patients to avoid unnecessary examinations.
